# Structural validity of the Rosenberg self-esteem scale in patients with schizophrenia in Indonesia

**Muhammad Muslih**[1,2], **Min-Huey Chung**[1,3]*

**1** School of Nursing, College of Nursing, Taipei Medical University, Taipei, Taiwan, **2** School of Nursing, Faculty of Health Science, Universitas Muhammadiyah Malang, Malang, Indonesia, **3** Department of Nursing, Shuang Ho Hospital, Taipei Medical University, New Taipei City, Taiwan

* minhuey300@tmu.edu.tw

## Abstract

### Background

The Rosenberg self-esteem scale (RSES) is a commonly employed instrument for measuring self-esteem in the general population and those with mental illness. However, confirmatory factor analyses (CFA) to determine the structural validity of the RSES for schizophrenia patients in Indonesia are limited.

### Objectives

We examined the structural validity of the RSES as a measurement for patients with schizophrenia in Indonesia through confirmatory factor analyses (CFA), as well as assessing internal consistency and reliability.

### Methods

The sample comprised 260 participants. Over two weeks, 30 subjects were added to investigate test-retest reliability. The structural validity analyzed was based on a CFA to determine the model fit. We used internal consistency (Ordinal alpha) to evaluate the reliability evidence.

### Results

Four different models were analyzed in this study. Considering the single-factor model (Model 1a), the overall fit criteria were inadequate. However, after some modification indices, all fit criteria were significantly adequate (Model 1b). The adequacy of all fit standards remained satisfactory when the two-factor model (Model 2) and hierarchical model (Model 3) were applied. The RSES had an ordinal alpha coefficient of 0.75. While 0.89 and 0.88 for the positive and negative self-esteem subscale, respectively. Test-retest reliability yielded adequate results with an interclass correlation score ranging from 0.87 to 0.93.

**Data Availability Statement:** All relevant data are within the paper and its S1 Dataset files. Please inform the authors if data are being used.

**Funding:** The authors did not received any funding for this study.

**Competing interests:** The authors declare that they have no conflicts of interest.

## Conclusions

The current investigation provided evidence supporting the structural validity, internal consistency, and reliability of the RSES, indicating that the RSES can be considered a valid and reliable measurement. A two-factor model of RSES was an appropriate model to measure self-esteem in our study. This finding suggests that the use of the RSES is beneficial and applicable in assessing levels of self-esteem in individuals diagnosed with schizophrenia in Indonesia.

## 1. Introduction

Self-esteem is an overall individual evaluation or appraisal of the self [1] and how a person thinks of themselves. Self-esteem is "the degree to which a person values, approves of, or likes himself or herself" [2]. Self-esteem is a crucial component of mental health and general psychological well-being. It influences an individual's achievements and successes, social interactions, and ability to cope with environmental stressors [3, 4]. Individuals with high self-esteem believe they possess many positive qualities and attitudes toward themselves [5]. In summary, self-esteem is a pivotal psychological construct [6] that controls several facets of an individual's existence, encompassing mental well-being, accomplishments, interpersonal engagements, and coping abilities [3, 4].

There is a reciprocal relationship between self-esteem and mental illnesses [7]. A previous study found that self-esteem plays a vital role in developing diverse mental illnesses and social problems encompassing a range of internalizing issues, such as depression, suicidal tendencies, eating disorders, and anxiety, as well as externalizing problems, including violence and substance abuse [8]. Conversely, it has been hypothesized that mental illnesses can lead to low self-esteem as a significant consequence [9]. Consistent with prior research, low self-esteem has been found to heighten susceptibility to the onset of mental illness [10]. Eventually, individuals with mental illnesses are likely to have fluctuating self-esteem levels [11]. Therefore, drawing from the aforementioned explanation, we can conclude that self-esteem is considered a component of self-assessment, which influences mental health and vice versa.

The Rosenberg Self-Esteem Scale (RSES), which was developed by Rosenberg [12], is one of the most extensively used instruments for measuring self-esteem globally [13–17]. Researchers often use the RSES to measure self-esteem in the clinical population, such as eating disorders [18], anxiety, depression [7], attention and emotional disorder [19], schizophrenia and bipolar disorder [20]. Other studies have tested the RSES in specific people, such as ex-prisoners [21], drug users [22], and single mothers [23]. The RSES has been translated and adapted into a number of different languages, including German [24], Dutch [25], Estonian [26], French [27], Portuguese [28], Spanish [29], Japanese [17], and Thai [14]; thus, making it applicable to participants from diverse samples or populations. It has been adapted across 53 nations with distinct ethnic groups and cultures [30]. This finding indicates that the RSES is widely used to measure self-esteem. Supporting this idea, a prior study suggests that the popularity of the RSES can be attributed to its brevity and simplicity, as it comprises only ten questions that can be completed within a short timeframe of 1 to 2 minutes [13].

Multiple countries, including Indonesia, have implemented the RSES to measure the self-esteem of college students and the general population [30]. Some previous studies have shown that self-esteem has been associated with schizophrenia [31–33]. For example, there was a

significant correlation between a decrease in the intensity of adverse symptoms and an enhancement in self-esteem, and conversely [34]. However, psychometric analyses, including structural validity and reliability, in patients diagnosed with schizophrenia were not explicitly addressed in previous studies [30, 35].

In a prior study, the factor structure of the RSES was examined using psychometric tests, and it focused on adolescents [36], as they were the original target population of this scale. It has also been tested in adults [37] and the general population [24]. Nevertheless, there is no available evidence supporting the utilization of RSES among individuals diagnosed with schizophrenia in Indonesia. Hence, evaluating the RSES in individuals diagnosed with schizophrenia is imperative to ascertain its psychometric analysis. This study aimed to assess the structural validity of the RSES as a measurement for patients with schizophrenia in Indonesia through confirmatory factor analyses (CFA), as well as assessing internal consistency and reliability.

## 2. Methods

### 2.1 Participants

This is an instrumental questionnaire validation study. Two psychiatric hospitals and one psychiatric rehabilitation center in East Java, Indonesia, were visited to obtain the required data. We distributed the questionnaire from August 2018 to February 2019. Participants were recruited using the convenience sampling technique. The following requirements had to be met for someone to be included: (a) they had been diagnosed with schizophrenia; (b) aged $\geq$ 20 years; (c) hospitalized in a psychiatric ward; and (d) able to speak, read, and write Indonesian. The Mini-Mental State Examination (MMSE) was utilized to screen out participants with cognitive impairment (i.e., MMSE scores < 24).

The size of 200 participants required in this study is acceptable based on recommendations from prior studies [38, 39]. In this study, data was missing from four questionnaires because they were not accurately completed, and 21 participants were excluded because their MMSE score was less than 24. Considering the response rate of 20%, the final sample comprised 260 participants, as presented in the (S1 Dataset). In addition, we recruited an additional 30 individuals to investigate test-retest reliability over two weeks.

### 2.2 Instruments

The study incorporates many socio-demographic data, including age as a continuous variable. The remaining variables as categorical variables, namely gender (male; female), marital status (single; married; divorced, or widowed), employment status (employed; unemployed), source of income (personal income; family support; personal and family support), education (elementary; junior; high school; university/ college), previous hospitalization (yes; no), and onset of illness (<1 year; 1–5 years; >5 years).

The RSES is not licensed and is available for public use. Information about the scale can easily be gathered, and permission to use this resource can be sought at https://socy.umd.edu/about-us/rosenberg-self-esteem-scale. The scale consists of 10 items evaluated on a 4-point Likert scale from 1 (strongly disagree) to 4 (strongly agree). Scores vary from 10 to 40, with higher scores indicating a more positive self-esteem appraisal. To measure the reverse score, five questions are worded positively (items 1, 3, 4, 7, and 10), while five are worded negatively (items 2, 5, 6, 8, and 9).

The validity and reliability evidence of the RSES were assessed in prior studies, and the obtained results are presented herein. Concerning the evaluation of construct validity, prior studies have demonstrated that the RSES yielded an excellent model fit [21, 40]. According to

a previous study, [41] it is suggested to ensure that the Cronbach alpha criteria for each sub-scale is ≥ 0.70. The Cronbach's alpha coefficients for the positive and negative self-esteem sub-scales were determined to be 0.96 and 0.98, respectively [21]. A prior study conducted on individuals who are native English speakers has also yielded Cronbach alpha coefficients of 0.87 and 0.75 for the subscales measuring positive and negative self-esteem, respectively [17].

## 2.3 Translation procedure

Due to their specificity and straightforwardness, we adhered to the parameters suggested by a previous study [42]. Initially, the original questionnaire was translated into Indonesian by two translators, a psychiatrist and a professional translator whose native language was Indonesian, with the author's approval. Both were bilingual and English-proficient. Second, we compared the two translated versions and created a new draft by combining the terminology and phrases supplied by the two translators in the previous step. Thirdly, the information was back-translated by two more independent translators with the same credentials and qualities as the first translators. Fourth, we compared the original questionnaire with the two back-translations of the questionnaire from the third phase. Considering the distance and time variations, we communicated with all four translators by email at this stage. Fifth, 30 volunteers were selected from a psychiatric hospital to evaluate the clarity of the questionnaire's instructions, items, and response format. In addition, we asked two professionals (a psychiatrist and a psychologist) for revisions and ideas. In the final phase, the full Indonesian version of the RSES scale was administered to the study sample and tested the evidence of validity and reliability (Fig 1).

## 2.4 Statistical analysis

This study utilized SPSS and AMOS version 23.0 software (IBM; Armonk, New York, USA) and R Studio version 4.3.2 (R Foundation, Vienna, Austria). All statistical significance was indicated by a $p$-value < 0.05.

**Descriptive analysis.** Descriptive statistics were used to present the demographic characteristics of the study. The proportion of participants who obtained minimum and maximum scores is defined as floor and ceiling effect, respectively. Continuous variables are presented as means and standard deviations, whereas categorical variables are expressed as frequencies and percentages. The quantitative characteristics of the RSES were computed as mean, standard deviation (SD), skewness, and kurtosis. A skewness value between −1 and 1 was considered adequate [43]. The degree of vertical spread in the mean distribution corresponded to the kurtosis. The normality was assumed if the kurtosis value was less than 2.5 times the standard error [44].

**Structural validity.** Structural validity pertains to the extent to which the scores of a Patient-Reported Outcome Measure (PROM) accurately represent the underlying structure of the construct being measured [41]. Assessment of structural validity is typically conducted through the use of CFA [41, 45]. A CFA was carried out to assess how well the RSES model fits the data. The following fit indices were utilized during the evaluation process: $X^2$/df, the comparative fit index (CFI), incremental fit index (IFI), the Tucker–Lewis index (TLI), goodness-of-fit index (GFI), adjusted goodness-of-fit index (AGFI), the standardized root mean square residual (SRMR), and the root mean square error of approximation (RMSEA). A chi-square with a degree of freedom ratio of less than 5.0 indicated that the model was a good fit [46]. An acceptable model fit was characterized by a GFI greater than 0.80 [47] and an AGFI of 0.80 to 0.90 [48–52]. When the CFI, IFI, and TLI values were all greater than 0.90 [53], the SRMR value was less than 0.08 [53, 54], and the RMSEA value was less than 0.10 [47], the model fit was deemed to be satisfactory. Akaike Information Criterion (AIC) and Bayesian Information Criterion (BIC) were used to evaluate the alternative model, with lower AIC and BIC values

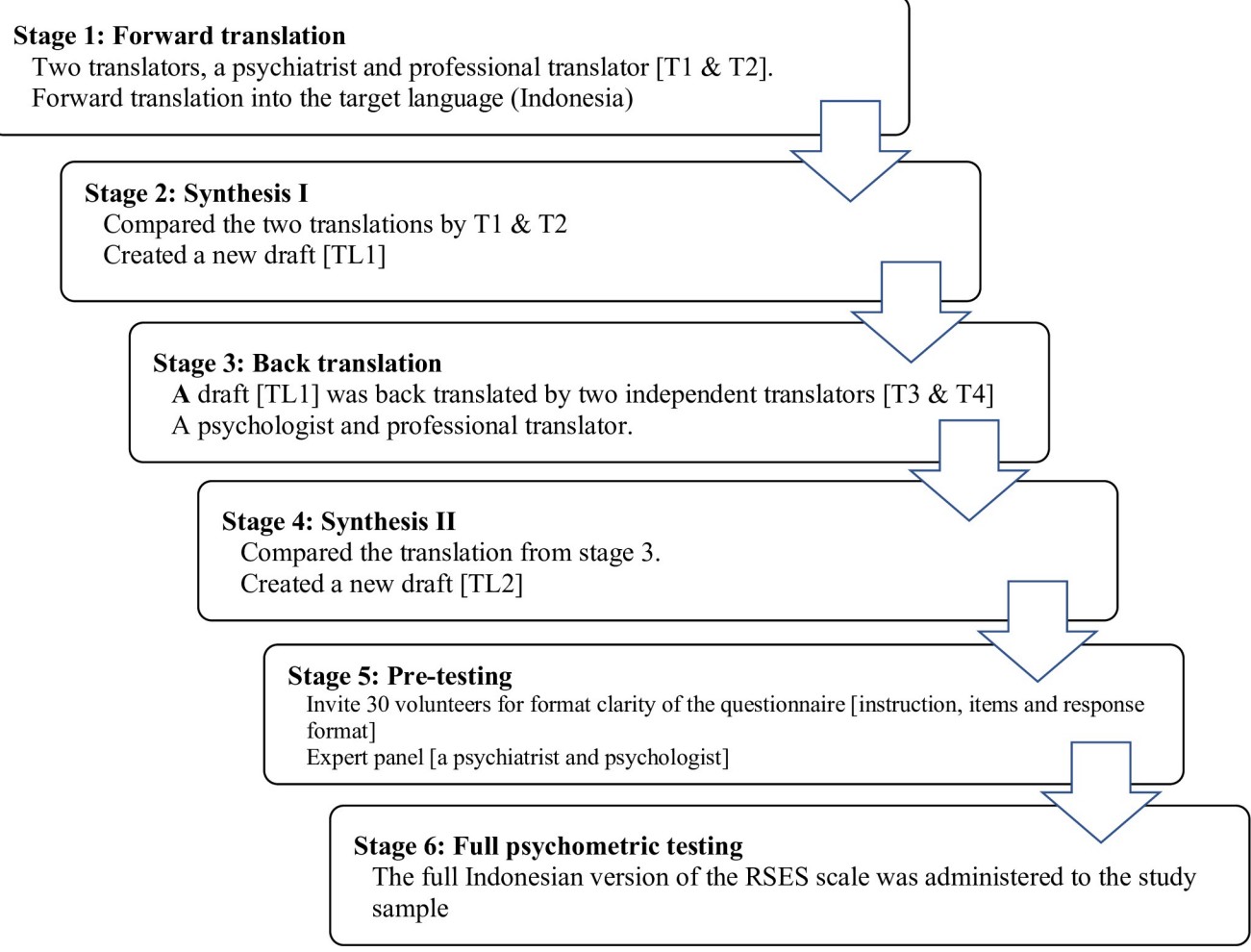

**Fig 1. Illustrates a flow diagram step-by-step parameters translation process of RSES, following the suggestions from the previous study.**

indicating the best model fit [55, 56]. The average variance extracted (AVE) is an integral component of the comprehensive construct validity assessment., and an AVE score greater than 0.50 indicates an adequate result [57].

**Internal consistency.** Internal consistency refers to "the extent of interrelatedness among the items and is commonly evaluated using Cronbach's alpha" [58]. Ordinal coefficient alpha is considered a viable alternative coefficient alpha to calculate a reliability estimate using the Likert response data [59, 60]. An alpha equal to or better than 0.70 demonstrates an adequate internal consistency [61, 62]. The composite reliability (CR) was used to measure a robust internal consistency. The values of CR greater than 0.70 indicated a significant result [63].

**Reliability.** Interclass correlation coefficient (ICC) was used to evaluate the reliability [41]. The ICC score between 0.75 and 0.90 indicated satisfactory reliability and consistency between two-time measurements, and a score greater than 0.90 revealed excellent reliability [64].

## 2.5 Ethical approval

The Ethics Research Committee approved this research of the University of Muhammadiyah Malang on July 19, 2018 (approval number: E.5.a/239/KEPK- UMM/VII/2018).

## 3. Results

Table 1 presents the clinical demographics of the study. The mean age of the 260 participants was 38.13 (SD = 9.56). Most participants were men (169, 65%) and single (139, 53.5%). Most were unemployed (173, 66.5%) and received financial support from their families (196, 75.4%). For 127 participants (48.8%), the highest educational level was senior high school. In total, 220 participants (84.6%) had previously been hospitalized, and the illness duration to onset was >5 years for 142 participants (54.6%).

Table 2 displays the descriptive statistics, inter-item correlation, and item-total correlation of the RSES. Item 8 and item 3 received the highest and lowest mean scores of 2.57 (SD = 0.87) and 1.92 (SD = 0.81), respectively. The skewness score ranged from 0.39 to 0.90 for the total RSES items, and the kurtosis score ranged from 0.74 to 0.75. The item-total correlation varied from 0.47 to 0.6. Floor and ceiling effect was found at 5.80% - 54.20%, respectively (Table 4).

**Table 1. Demographic characteristics of the study (*n* = 260).**

| Characteristics | Participants (*n* = 260) | |
|---|---|---|
| | **Mean (SD)** | **n (%)** |
| Age | 38.13 (9.56) | |
| Gender | | |
| Male | | 169 (65.00) |
| Female | | 91 (35.50) |
| Marital status | | |
| Single | | 139 (53.46) |
| Married | | 81 (31.15) |
| Divorce or widowed | | 40 (15.39) |
| Employment status | | |
| Employed | | 87 (33.50) |
| Unemployed | | 173 (66.50) |
| Source of income | | |
| Personal income | | 30 (11.50) |
| Family support | | 196 (75.40) |
| Personal and family | | 34 (13.10) |
| Education | | |
| Elementary school | | 63 (24.23) |
| Junior high school | | 44 (16.92) |
| Senior high school | | 127 (48.85) |
| University/college | | 26 (10.00) |
| Previous hospitalization | | |
| Yes | | 220 (84.60) |
| No | | 40 (15.40) |
| Onset of illness | | |
| <1 year | | 75 (28.85) |
| 1–5 years | | 43 (16.54) |
| >5 years | | 142 (54.61) |

*missing data = 4

**MMSE score < 24 = 21

SD = standard deviation; MMSE = Mini-Mental State Examination.

**Table 2. Descriptive statistics, interitem, and item-total correlation of the RSES items ($n = 260$).**

| Items | | Inter-item correlation ($n = 260$) | | | | | | | | | |
|---|---|---|---|---|---|---|---|---|---|---|---|
| | | 1 | 2 | 3 | 4 | 5 | 6 | 7 | 8 | 9 | 10 |
| RSES 1 | On the whole, I am satisfied with myself. | 1 | | | | | | | | | |
| RSES 2 | At times, I think I am no good at all. | −0.11 | 1 | | | | | | | | |
| RSES 3 | I feel that I have a number of good qualities. | 0.56 | −0.12 | 1 | | | | | | | |
| RSES 4 | I am able to do things as well as most other people. | 0.51 | −0.04 | 0.63 | 1 | | | | | | |
| RSES 5 | I feel I do not have much to be proud of. | −0.03 | 0.48 | −0.13 | −0.11 | 1 | | | | | |
| RSES 6 | I certainly feel useless at times. | −0.00 | 0.62 | −0.04 | −0.01 | 0.48 | 1 | | | | |
| RSES 7 | I feel that I'm a person of worth, at least on an equal plane with others. | 0.52 | −0.01 | 0.64 | 0.86 | −0.07 | 0.06 | 1 | | | |
| RSES 8 | I wish I could have more respect for myself. | −0.19 | 0.71 | −0.22 | −0.15 | 0.43 | 0.50 | −0.15 | 1 | | |
| RSES 9 | All in all, I am inclined to feel that I am a failure. | −0.1 | 0.85 | −0.13 | −0.09 | 0.56 | 0.63 | −0.05 | 0.71 | 1 | |
| RSES 10 | I take a positive attitude toward myself. | 0.55 | 0.01 | 0.58 | 0.63 | −0.01 | −0.03 | 0.64 | −0.08 | −0.04 | 1 |
| Item-total correlation | | **0.49** | **0.61** | **0.50** | **0.58** | **0.47** | **0.59** | **0.62** | **0.46** | **0.61** | **0.59** |
| Mean | | 2.05 | 2.39 | 1.92 | 1.98 | 2.32 | 2.30 | 2.02 | 2.57 | 2.31 | 1.97 |
| SD | | 0.94 | 0.86 | 0.81 | 0.84 | 0.86 | 0.91 | 0.83 | 0.87 | 0.83 | 0.86 |
| Skewness | | 0.58 | −0.02 | 0.90 | 0.69 | 0.32 | 0.21 | 0.59 | −0.39 | 0.28 | 0.72 |
| Kurtosis | | −0.55 | −0.68 | 0.75 | 0.05 | −0.47 | −0.74 | −0.08 | −0.53 | −0.40 | 0.00 |

SD = standard deviation

## 3.1 Structural validity

The goodness of indices for all alternative models is shown in Table 3. The AVE values were 0.69 and 0.68, and the square roots of the AVE were 0.83 and 0.82, indicating that each measured variable was significant (Table 4) (Figs 2–5).

## 3.2 Internal consistency and reliability evidence

The overall score of the RSES has an alpha coefficient of 0.75, as determined by the ordinal alpha approach. The ordinal alpha results for each subscale were 0.89 and 0.88, demonstrating

**Table 3. Goodness of fit indexes of the RSES Indonesian version.**

| Item | Model 1 | Model 2 | Model 3 | Model 4 |
|---|---|---|---|---|
| $X^2$/df | 24.50 | 2.96 | 2.93 | 2.93 |
| GFI | 0.55 | 0.93 | 0.93 | 0.93 |
| AGFI | 0.30 | 0.88 | 0.88 | 0.88 |
| CFI | 0.48 | 0.97 | 0.96 | 0.96 |
| TLI | 0.33 | 0.94 | 0.95 | 0.95 |
| IFI | 0.48 | 0.97 | 0.96 | 0.96 |
| SRMR | 0.19 | 0.03 | 0.04 | 0.04 |
| RMSEA | 0.30 | 0.08 | 0.08 | 0.08 |
| AIC | 897.634 | 133.992 | 141.497 | 141.497 |
| BIC | 968.848 | 240.812 | 216.27 | 216.27 |

df = degree of freedom GFI = goodness of fit index; AGFI = adjusted goodness of fit index; CFI = comparative fit index; TLI = Tucker-Lewis index; IFI = incremental fit index; SRMR = standardized root mean square residual; RMSEA = root mean square error of approximation; AIC = Akaike information criterion; BIC = Bayesian information criterion

**Table 4. Reliability analysis for the Indonesian adaptation of the rosenberg self-esteem scale (*n* = 260).**

| Items | % floor effect | % ceiling effect | Ordinal alpha | Test-retest reliability | | AVE | CR |
|---|---|---|---|---|---|---|---|
| | | | | ICC | 95% CI | | |
| Positive self- esteem | | | 0.89 | 0.93 | 0.88–0.96 | 0.69 | 0.92 |
| RSES 1 | 9.20 | 39.60 | | | | | |
| RSES 3 | 6.50 | 54.20 | | | | | |
| RSES 4 | 6.50 | 48.80 | | | | | |
| RSES 7 | 5.80 | 48.80 | | | | | |
| RSES 10 | 6.90 | 46.90 | | | | | |
| Negative self-esteem | | | 0.88 | 0.87 | 0.78–0.93 | 0.68 | 0.91 |
| RSES 2 | 8.50 | 38.10 | | | | | |
| RSES 5 | 10.40 | 47.30 | | | | | |
| RSES 6 | 10.40 | 40.00 | | | | | |
| RSES 8 | 10.80 | 50.00 | | | | | |
| RSES 9 | 8.50 | 47.70 | | | | | |

CI = confidence interval, ICC = intraclass correlation coefficient, AVE = average variance extracted, CR = composite reliability

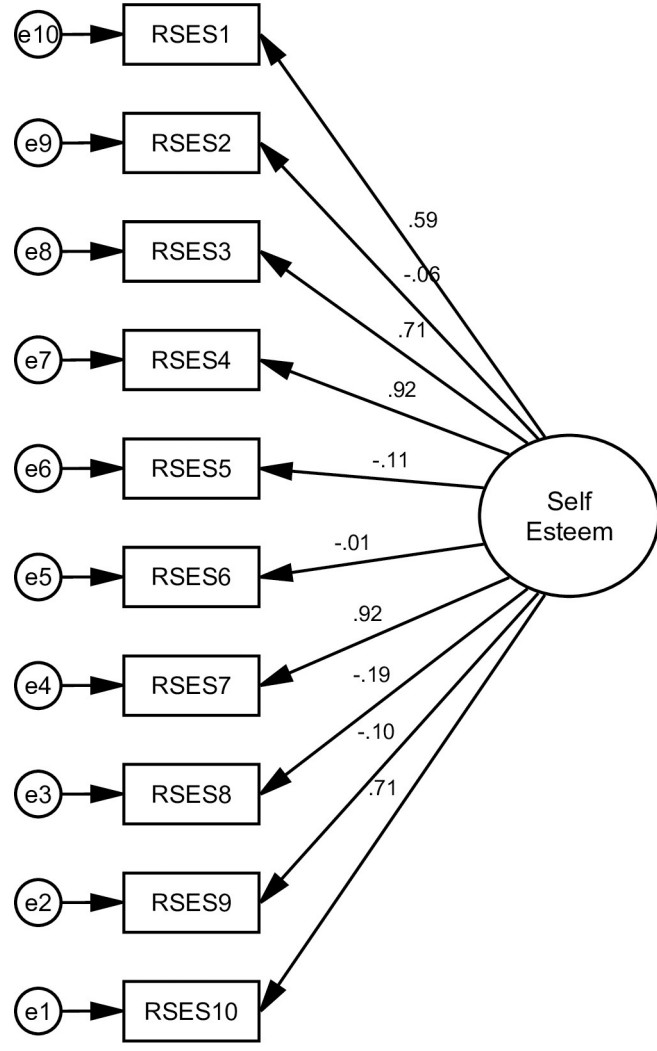

**Fig 2. Shows a single-factor model of the RSES.** This model extracted all items of RSES in the single-factor or uni-dimensional model (Model 1). However, the overall fit criteria were inadequate.

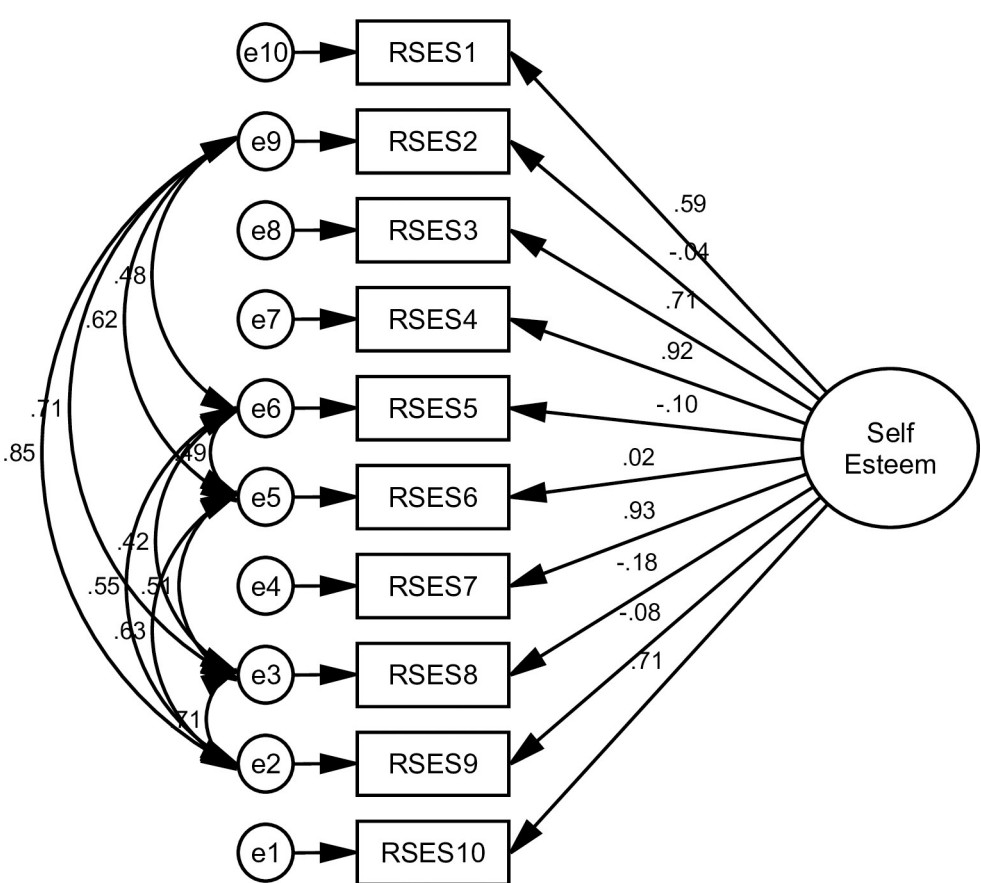

**Fig 3. Shows a single-factor model with correlated error.** Some modification indices were applied to Model 1. It was utilized to enhance the model fit. The result shows that all fit criteria were significantly adequate (Model 2).

acceptable internal consistency. As presented in Table 4, the CR was calculated for positive and negative factors, and the values were 0.92 and 0.91, respectively. Test-retest reliability exhibited satisfactory results, with an ICC between 0.87 and 0.93 (Table 4).

## 4. Discussion

This study aimed to establish the structural validity of the RSES as well as assess internal consistency and reliability. We included patients with schizophrenia in our study, which was not the case in the previous study [30]. The sample size of our study was adequate to perform factor analysis. In addition, the structural validity of the RSES was demonstrated through CFA, an approach that has not been conducted in prior studies in Indonesia.

Through the CFA, we examine four models of structural validity in our study, which are; the single-factor or uni-dimensional model (M1), the single-factor with correlated error (M2), the two-factor model (M3), and the hierarchical model (M4). The findings from M1, suggest that there is insufficient data to support the acceptance of a single factor. Where every model fit criterion failed to meet the required levels. A notable distinction was observed while implementing the correlated error in M2, demonstrating an enhancement in the good fit criteria. This adjustment was done on the negatively worded item. This finding aligns with some prior studies that have examined the presence of method effects related to negative items on the RSES [65–67]. While some of the model fit criteria are met, it is unfortunate that this model

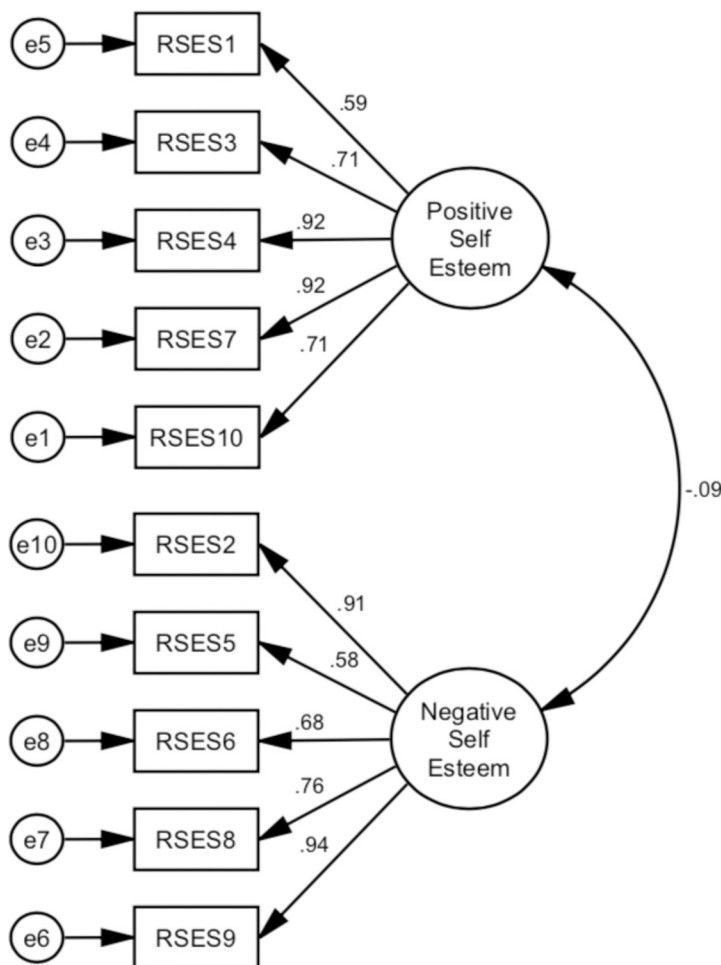

**Fig 4. Illustrates a two-factor model (Model 3) that divides RSES into two different factors, namely positive and negative self-esteem respectively.** All fit criteria remained adequate in this model.

also fell outside of other criteria. The BIC value is significantly higher compared to M3 and M4. Following a previous study, the difference in BIC scores greater than 10 provides strong evidence of the model [68]. If the objective is to achieve a goodness of fit, the BIC is the preferred option. Therefore, using BIC is more advantageous when selecting an accurate model [69].

The adequate fit indexes were also obtained in M3 and M4. The two-factor and hierarchical models exhibit comparable model fit in their respective analyses. Based on the findings mentioned above, it is suggested that the RSES can be characterized as two factors, which are positive and negative self-esteem. A previous study also referred to these two factors as positive and negative self-esteem [70]. The influence of wording effect on scale items may result in or contribute to a two-factor model. The wording item effect, then further related to the method effect has been observed in earlier studies [37, 71], which suggests the presence of a two-factor of RSES [71]. In summary, our finding indicates that the RSES scale has an acceptable model fit with two factors. Similar findings were also demonstrated in a previous study that a two-factor model was deemed to have an adequate model fit [21, 36, 72, 73]. Based on our findings, we can conclude that the RSES which is a two-factor model was a valid instrument for people with schizophrenia in Indonesia. Acknowledging the necessity of reassessing the utilization of

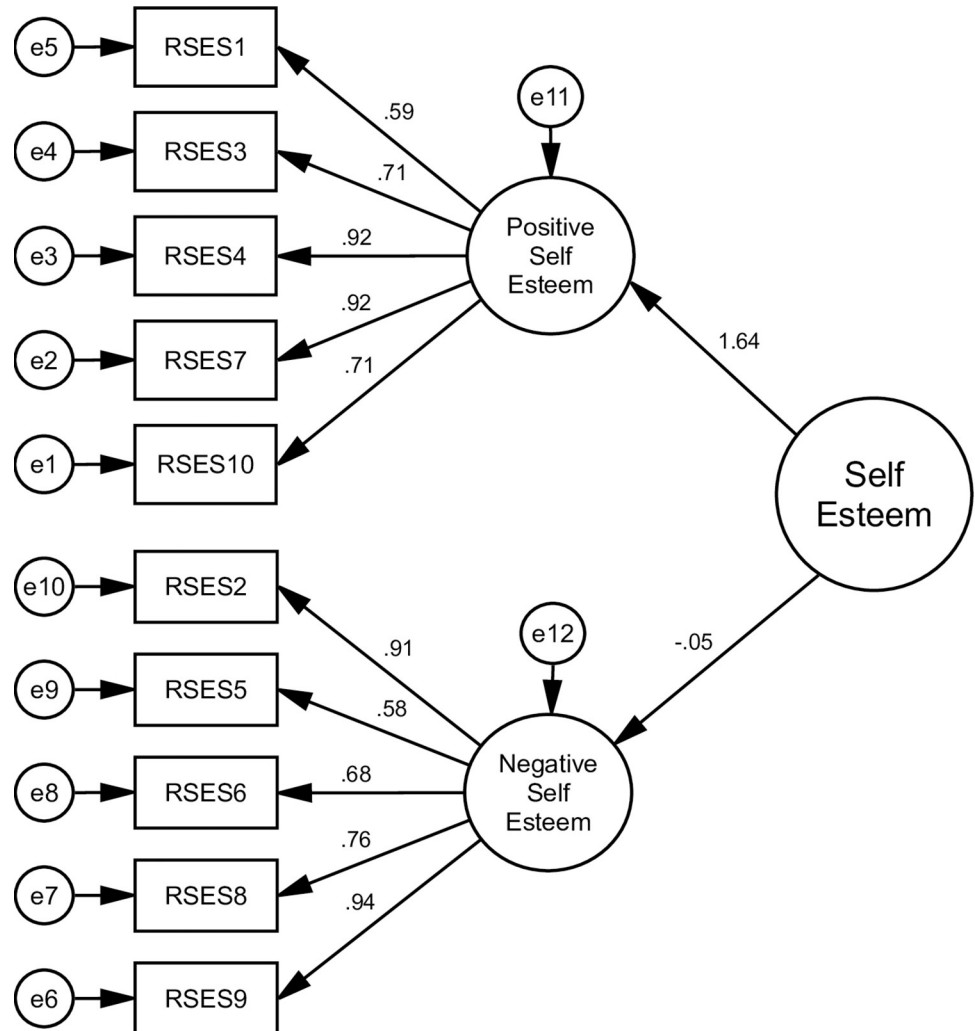

**Fig 5. Presents the hierarchical factor model as a second-order model (Model 4).** Adequate fit indexes have been found in this model, similar to model 3. Henceforth, these four models were referred to as M1, M2, M3, and M4 respectively.

the RSES and its theoretical foundations in administering the scale to target populations is essential.

The total score on the Indonesian version of the RSES had an ordinal alpha coefficient of 0.75. In contrast, the positive and negative self-esteem subscale had a coefficient of 0.89, and 0.88 respectively. In line with this finding, a previous investigation found satisfactory levels of internal consistency, as measured by a Cronbach's alpha coefficient that ranged from 0.81 to 0.91 [13, 14, 25, 30]. Our result was consistent with a previous study [74], which was conducted in individuals with severe mental illnesses, not specific only to patients with schizophrenia, and which reported strong internal consistency. To demonstrate more robust internal consistency, composite reliability also exhibits favorable outcomes.

In addition, we looked at the test-retest reliability of the RSES. The ICC values generated were 0.93 and 0.87 for the positive and negative self-esteem respectively. The ICC results were adequate, indicating the stability of each factor of the RSES. Additionally, it was observed that there was a high correlation coefficient between the test-retest reliability and Cronbach's

alpha. In conclusion, the findings of this study provide evidence supporting the robust reliability of the RSES as a reliable instrument for assessing self-esteem in individuals diagnosed with schizophrenia in Indonesia.

The present study also has limitations. Because our study sample included only patients with schizophrenia, these findings cannot be extrapolated to populations with other mental illnesses. Moreover, most of our study's participants are men, leading to a more prominent interpretation of item scores among this group. Future studies should aim to expand the participant pool by including individuals diagnosed with a diverse range of mental illnesses while also ensuring a balanced representation of both genders. In addition to the limitations mentioned above, it is essential to note that our study produced favorable evidence regarding structural validity, internal consistency, and reliability. Hence establishing the RSES as a valid and reliable questionnaire appropriate for implementation within the tested sample group.

## 5. Conclusion

The current investigation provided evidence supporting the structural validity, internal consistency, and reliability of the RSES, indicating that the RSES can be considered a valid and reliable measurement. A two-factor model of RSES was an appropriate model to measure self-esteem in our study. This finding suggests that the use of the RSES is beneficial and applicable in assessing levels of self-esteem in individuals diagnosed with schizophrenia in Indonesia. Nevertheless, further research is needed to understand better the characteristics of the method factors for different populations. As previously stated, positive and negative item wording is still a major consideration in psychometric analysis.

## Supporting information

**S1 Dataset.**
(XLSX)

## Acknowledgments

We would like to acknowledge Wallace Academic Editing Company for editing this manuscript.

## Author Contributions

**Data curation:** Muhammad Muslih.

**Formal analysis:** Muhammad Muslih, Min-Huey Chung.

**Supervision:** Min-Huey Chung.

**Writing – original draft:** Muhammad Muslih.

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
