## [Decision Letter · Decision Letter 0]

25 May 2023

PONE-D-23-08354

Construct Validity of the Rosenberg Self-Esteem Scale in patients with Schizophrenia in Indonesia

PLOS ONE

Dear Dr. Chung,

Thank you for submitting your manuscript to PLOS ONE. After careful consideration, we recommend reconsideration of your manuscript following major revision. Therefore, we invite you to submit a revised version of the manuscript that addresses the points raised during the review process.

ACADEMIC EDITOR:Please respond to and address all the reviewers' recommendations before resubmission. The reviewers have pointed out that the introduction and methodology sections of the paper require improvement. Specifically, a sample of 260 subjects was used to perform an EFA and CFA.  It is crucial to take their suggestions and make the necessary revisions to ensure that the paper complies with quality standards of the instruments validation. 

We look forward to receiving your revised manuscript.

Kind regards,

Silvia Escribano Cubas

Academic Editor

PLOS ONE

Reviewers' comments:

Reviewer's Responses to Questions

**Comments to the Author**

1. Is the manuscript technically sound, and do the data support the conclusions?

Reviewer #1: Yes

Reviewer #2: Partly

Reviewer #3: Yes

2. Has the statistical analysis been performed appropriately and rigorously? 

Reviewer #1: Yes

Reviewer #2: No

Reviewer #3: Yes

3. Have the authors made all data underlying the findings in their manuscript fully available?

Reviewer #1: No

Reviewer #2: Yes

Reviewer #3: Yes

4. Is the manuscript presented in an intelligible fashion and written in standard English?

Reviewer #1: No

Reviewer #2: No

Reviewer #3: Yes

5. Review Comments to the Author

**Reviewer #1:** Review of PONE-D-23-08354

The submission studies the psychometric properties of an Indonesian version of the Rosenberg Self-Esteem Scale using a sample of inpatients diagnosed with schizophrenia. The Introduction is brief and does not justify the approach of the study sufficiently. Some sentences are confusing, e.g., in the first paragraph when the authors state that SE plays a role in the development of mental illness (i.e., precedent) but also state that this is because SE is a key outcome of mental illnesses (i.e., subsequent).

Some results are confusing, e.g., using the same sample to conduct first an EFA and second a CFA, or describing the psychometric properties as validity. As a reader, I would expect reports of associations with other constructs typically related to SE and some sort of criterion validity. The authors also discuss the two-factor model without acknowledging previous studies on factor-solutions in relation to wording. What is however an interesting finding is the extremely small association between the positive and the negative factor which is not in line with previous research. It might be beneficial to study this further.

The sample is interesting and the psychometric test plausible, but I would expect such tests in such a sample as a first step of a paper on further relevant research questions. The psychometric tests alone do not seem like a substantial contribution to the literature.

**Reviewer #2: **

ABSTRACT:

-The use of some terms when mentioning types of validity evidence, do not follow the current Standards. For example, the term "construct validity" should be replaced, according to the last versions of the Standards (AERA, APA, & NCME, 1999, 2014).

INTRODUCTION:

-"According to literature [6], self-esteem also plays a crucial role in developing various mental illnesses" Please determine the type and characteristics of mental illnesses.

-The use of some terms when mentioning types of validity evidence, do not follow the current Standards. For example, the term "construct validity" should be replaced, according to the last versions of the Standards (AERA, APA, & NCME, 1999, 2014).

METHODS:

-"This study employed a cross-sectional design" Please determine the correct type of study. This is an instrumental questionnaire validation study, not a cross-sectional study.

-"(e) provision of informed consent"The following element cannot be considered an inclusion criterion. This aspect should be considered in the ethics section.

-"The participants were informed before signing the written consent form that participation is voluntary and their confidentiality was carefully guarded. The purposes of the study, procedure for filling out questionnaires, potential risk or inconvenient, confidentiality issues and contact detail of the researcher were explained in the written consent forms and informs the participant directly. The researcher also gives an option to the participants to not participate in the study. The participants in this study was vulnerable group (psychiatric patients/ case with mental illness), therefore, additional protection was provided"  Please review the flow of information in the methods section. From my humble point of view there are certain aspects that are not in the corresponding section (in this case ethics), besides repeating information.

-"The minimum number of participants recommended by Mundfrom, Shaw [29] was 20 times the number of variables, which in this case equaled 200 participants. In this study, data was missing from four questionnaires because they were not accurately completed, and 21 participants were excluded because their MMSE score was less than 24. Considering the response rate of 20%, the final sample comprised 260 participants" I seem to remember that the rule of 5 and 20 participants has long been out of use, it is calculated on the basis of expected communalities and items per factor, taking into account the work in statistical analysis under optimal conditions. I refer the authors to the citations of Ferrando-Piera et al. (2022) and Lloret-Segura et al. (2014).

-"Sinclair, Blais [9] found that the dependability of the RSES was satisfactory, with a Cronbach's alpha of 0.91" Please determine internal consistency by scales or sub-dimensions, currently presenting internal consistency of total scale scores is a major methodological error. I refer the authors to the citation of Prinsen et al. (2018).

- "Franck, De Raedt [21] tested the construct validity by correlating the RSES score with the NEO Five-Factor Inventory subscales. In line with Tinakon and Nahathai [10], the RSES negatively correlated with the Thai Depression Inventory"  Please determine the magnitude of the correlations in the different studies. The use of some terms when mentioning types of validity evidence, do not follow the current Standards.

-"In the last phase, the complete Indonesian version of the RSES scale was administered to the study sample, and its validity and reliability were determined"  The use of some terms when mentioning types of validity evidence, do not follow the current Standards.

-"A Cronbach's alpha equal to or better than 0.70 demonstrates adequate internal consistency [33, 34]"  From my humble point of view the authors should have calculated the ordinal alpha. The rest of the statistics in categorical response scales underestimate reliability. I refer the authors to the citation of Zumbo et al. (2007).

- "We carried out an EFA (exploratory factor analysis) to examine the construct validity of the scale using a principal component analysis [36, 37] with varimax rotation [38]" I seem to remember that the use of orthogonal rotations in highly correlated multidimensional constructs may deform the underlying structure of test item scores. I refer the authors to the citations of Ferrando-Piera et al. (2022), Lloret-Segura et al. (2014) and Prinsen et al. (2018).

- "to determine the number of components, an eigenvalue greater than 1.00 was utilized [43, 44]"  According to the latest standards it is more correct to use parallel analysis to determine the number of factors. I refer the authors to the citations of Ferrando-Piera et al. (2022) and Lloret-Segura et al. (2014). Adequacy of EFA (KMO; Barteltt's test of sphericity and determinant) should come before "parallel analysis" or "number of factors" to determine the number of factors to be retained, not after as now appears in several places of the text.

-I do not observe the criteria used to retain or discard items in the EFA; the authors should provide scientific evidence in this regard.

-The authors do not make it clear whether they randomly segmented the sample into two halves to perform the CFA and EFA.

-Why do the authors perform an EFA and a CFA with the same sample, what does this contribute to the study?

-I still do not understand why with a sample of 260 participants and without segmenting it the authors perform an EFA and a CFA, taking into consideration that a sample of 130 to perform the different statistical methods would be unstable due to the small size. I refer the authors to the citations of Ferrando-Piera et al. (2022) and Lloret-Segura et al. (2014).

-From my humble point of view it would be more interesting to perform only a CFA with several adjustments: a) congeneric model, b) correlated error model if the modification rates are greater than or equal to 35000 and c) a second order model to assess the existence of a single underlying construct.

-"convergent validity" The use of some terms when mentioning types of validity evidence, do not follow the current Standards. I refer the authors to the citation of Prinsen et al. (2018).

RESULTS:

-There are variables that have not been described in the methodology section. The authors should include a section where readers can see all the sociodemographic variables collected. The authors should write the variables more clearly, specifically if they are continuous or categorical variables. In the case of categorical variables, they should include the response options in parentheses. The authors should describe the evaluation instruments more clearly. I recommend writing it with a hyphen, the name of the instrument and its psychometric properties.

-The authors should provide normative and performance data for the scale items (ceiling and floor effects). Based on the results found , if floor and ceiling effects are found, the authors should provide percentiles to facilitate the interpretation of the scores. Authors should explore differences between men and women and provide effect sizes. However, t-test for examining differences in direct score between ratings of men and women should be conducted after demonstrating measurement invariance across this variable, at least at the factor loadings level (weak o metric invariance) and item thresholds/intercepts level (strong or scalar invariance), to ensure that direct scores are operating in the same way and, therefore, are meaningfully comparable.

DISCUSSION:

-"In addition, evidence of the construct validity of the RSES was demonstrated through EFA and CFA, contrary to previous studies conducted in Indonesia"  Is it appropriate to do this with a total sample of 260 participants?

-"In line with this finding, a previous investigation found satisfactory levels of internal consistency, as measured by a Cronbach's alpha coefficient that ranged from 0.81 to 0.91 [9, 10, 21, 25]"  What are the implications of obtaining an internal consistency greater than 0.90?

-"Another study indicated that a two-factor model considerably increases the fit compared to a one-factor model [58]" The following statement supports my humble opinion of performing a CFA with several adjustments, specifically congeneric and a second-order model.

-Authors should include in the limitations section that the majority of the sample on which we can interpret the item scores are men.

CONCLUSIONS:

-The authors' statements in the conclusions section should be based on the statistical recommendations provided. Are we sure that the structure has not been influenced by orthogonal rotation?

ADDITIONAL COMMENTS:

-"Table 1. Demographic Characteristics of the Study (n = 260)"  The sum of the percentages exceeds or does not reach 100%, the authors should check the sum of the tables.

BIBLIOGRAPHY:

Ferrando-Piera, P. J., Lorenzo-Seva, U., Hernández-Dorado, A., & Muñiz-Fernández, J. (2022). Decálogo para el Análisis Factorial de los Ítems de un Test. Psicothema. Retrieved from https://hdl.handle.net/11162/217964.

Lloret-Segura, S., Ferreres-Traver, A., Hernández-Baeza, A., & Tomás-Marco, I. (2014). El análisis factorial exploratorio de los ítems: una guía práctica, revisada y actualizada. Psicothema. https://doi.org/6018/analesps.30.3.199361

Prinsen, C. A., Mokkink, L. B., Bouter, L. M., Alonso, J., Patrick, D. L., De Vet, H. C., & Terwee, C.

B. (2018). COSMIN guideline for systematic reviews of patient-reported outcome measures. Quality of life research, 27(5), 1147-1157. https://doi.org/10.1007/S11136-018-1798-3

Zumbo, B.D., Gadermann, A.M., & Zeisser, C. (2007). Ordinal versions of coecients alpha and theta for Likert rating scales. Journal of Modern Applied Statistical Methods,6, 21-29. https://doi.org/10.22237/jmasm/1177992180

**Reviewer #3: **

It is a ver straightforward article in which the construct validity of the Rosenberg Self-Esteem Scale for schizophrenia patients in Indonesia is assessed.

I have to admit that I was not expecting the length of this article and I received it with great pleasure. Nowadays articles this long are not that easy to find. It is very easy to read and very straightforward as well, which is one of its strong points, in my humble opinion.

However, the introduction section is lacking some depth. Authors state that "self-steem plays a crucial role in developing various mental illnesses" (page 3, line 7). This affirmation makes me think that self-steem triggers some mental disorders. But then, following the previous sentence: "this is because low self-esteem has been postulated to be one of the primary outcomes of mental illness". This part needs some clarification. Also, in the last paragraph of the introduction (page 4, line 2), authors state: "Additionally, some previous studies have shown that self-esteem has been associated with schizophrenia". In which way? In general, although I can guess the intention of the authors of making a short paper, I think the introduction section would benefit from a deeper research.

Methods, results and discussion sections are great. They are short and straightforward, like the rest of the paper.

My main concern is the fact that references do not follow the APA format. Please revise it carefully.

---

## [Author Response · Author response to Decision Letter 0]

4 Oct 2023

Dear Editor and Reviewer

Manuscript ID: 

Manuscript Title: “Construct Validity of the Rosenberg Self-Esteem Scale in Patients with Schizophrenia in Indonesia”

We sincerely thank the editor and all reviewers for their valuable suggestions and for giving us a great opportunity to revise our manuscript entitled “Construct Validity of the Rosenberg Self-Esteem Scale in Patients with Schizophrenia in Indonesia”. We have incorporated all the suggested changes into the manuscript and have highlighted the revised sections. Hereby, our responses and revisions are based on the editor and reviewer’s comments.

EDITORIAL COMMENTS:

Please respond to and address all the reviewers' recommendations before resubmission. The reviewers have pointed out that the introduction and methodology sections of the paper require improvement. Specifically, a sample of 260 subjects was used to perform an EFA and CFA. It is crucial to take their suggestions and make the necessary revisions to ensure that the paper complies with the quality standards of the instrument validation.

Response: Thank you for your correction. The necessary amendments and recommendations provided by the reviewers have been incorporated into our work.

REVIEWER #1 COMMENTS:

1. The submission studies the psychometric properties of an Indonesian version of the Rosenberg Self-Esteem Scale using a sample of inpatients diagnosed with schizophrenia. The Introduction is brief and does not justify the approach of the study sufficiently. Some sentences are confusing, e.g., in the first paragraph when the authors state that SE plays a role in the development of mental illness (i.e., precedent) but also state that this is because SE is a key outcome of mental illnesses (i.e., subsequent).

Response: Thank you for your corrections. We want to explain that self-esteem and mental illnesses have a bidirectional pattern. Low self-esteem makes people more vulnerable to the development of mental illnesses, and mental illness, in turn, lowers self-esteem. The paragraph has been revised to enhance the reader's comprehension of the author's intended message. 

(Page 3)

There exists a reciprocal relationship between self-esteem and mental illnesses. A previous study found that self-esteem plays a pivotal role in developing diverse mental illnesses and social problems encompassing a range of internalizing issues, such as depression, suicidal tendencies, eating disorders, and anxiety, as well as externalizing problems, including violence and substance abuse [6]. Conversely, it has been hypothesized that mental illnesses can lead to low self-esteem as a significant consequence [7]. Consistent with prior research [8] low self-esteem has been found to heighten susceptibility to the onset of mental illness. In contrast, the existence of mental illness subsequently diminishes self-esteem. Eventually, individuals with mental illnesses are likely to have fluctuating self-esteem levels. Hence, self-esteem is considered a component of self-assessment, which influences mental health and vice versa.

2. Some results are confusing, e.g., using the same sample to conduct first an EFA and second a CFA, or describing the psychometric properties as validity. As a reader, I would expect reports of associations with other constructs typically related to SE and some sort of criterion validity. The authors also discuss the two-factor model without acknowledging previous studies on factor solutions in relation to wording. What is however an interesting finding is the extremely small association between the positive and the negative factor which is not in line with previous research. It might be beneficial to study this further.

Response: Thank you for your corrections. We have made the necessary revisions to our work. As a result, we have decided to exclusively focus on presenting confirmatory factor analysis (CFA) in order to demonstrate validity evidence of the RSES. (Statistical analysis; Page 7); (Results; Page 8); (Discussion; Page 9).

3. The sample is interesting and the psychometric test plausible, but I would expect such tests in such a sample as a first step of a paper on further relevant research questions. The psychometric tests alone do not seem like a substantial contribution to the literature.

Response: Thank you for your corrections. As previously stated, the RSES has been widely used as a scale to measure self-esteem in a variety of populations; but, to the best of our knowledge, there has been no previous research employing the RSES in schizophrenia patients, particularly in Indonesia. As a result of our hypothesis, the RSES psychometric test in our study can make a significant contribution. In addition, we revised and added more information in the last paragraph of the introduction section to describe our novelties. 

(Page 4)

Multiple countries, including Indonesia, have implemented the RSES to measure the self-esteem of college students and the general population [26]. Interestingly, some previous studies have shown that self-esteem has been associated with schizophrenia [27-29]. For example, there was a significant correlation between a decrease in the intensity of adverse symptoms and an enhancement in self-esteem, and conversely, [30]. Unfortunately, the scale for people diagnosed with schizophrenia was not evaluated using psychometric testing in a prior study [26]. Additionally, they only focused on internal consistency and factor structure invariance.

 

REVIEWER #2 COMMENTS:

ABSTRACT:

1. The use of some terms when mentioning types of validity evidence, do not follow the current Standards. For example, the term "construct validity" should be replaced, according to the last versions of the Standards (AERA, APA, & NCME, 1999, 2014).

Response: Thank you for your corrections. We have corrected and changed it following the current standards.

(Page 2)

We examined the validity and reliability evidence of the RSES by measuring confirmatory factors among patients with schizophrenia in Indonesia

INTRODUCTION:

2. According to literature [6], self-esteem also plays a crucial role in developing various mental illnesses" Please determine the type and characteristics of mental illnesses.

Response: Thank you for your corrections. According to the intended literature, we amended and added information about the types and characteristics of mental illnesses.

(Page 3)

There exists a reciprocal relationship between self-esteem and mental illnesses. A previous study found that self-esteem plays a pivotal role in developing diverse mental illnesses and social problems encompassing a range of internalizing issues, such as depression, suicidal tendencies, eating disorders, and anxiety, as well as externalizing problems, including violence and substance abuse [6]. Conversely, it has been hypothesized that mental illnesses can lead to low self-esteem as a significant consequence [7]. Consistent with prior research [8] low self-esteem has been found to heighten susceptibility to the onset of mental illness. In contrast, the existence of mental illness subsequently diminishes self-esteem. Eventually, individuals with mental illnesses are likely to have fluctuating self-esteem levels. Hence, self-esteem is considered a component of self-assessment, which influences mental health and vice versa.

3. The use of some terms when mentioning types of validity evidence, do not follow the current Standards. For example, the term "construct validity" should be replaced, according to the last versions of the Standards (AERA, APA, & NCME, 1999, 2014).

Response: Thank you for your corrections. We have corrected and changed it following the current standards.

(Page 4)

Hence, evaluating the RSES in individuals diagnosed with schizophrenia is imperative to ascertain its validity and reliability

METHODS:

4. This study employed a cross-sectional design" Please determine the correct type of study. This is an instrumental questionnaire validation study, not a cross-sectional study.

Response: Thank you for your correction. We have corrected and replaced it in accordance with your suggestions.

(Page 5)

This is an instrumental questionnaire validation study. 

5. (e) provision of informed consent"The following element cannot be considered an inclusion criterion. This aspect should be considered in the ethics section.

Response: Thank you for your correction. We have corrected it in accordance with your suggestions.

6. The participants were informed before signing the written consent form that participation is voluntary and their confidentiality was carefully guarded. The purposes of the study, procedure for filling out questionnaires, potential risk or inconvenient, confidentiality issues and contact detail of the researcher were explained in the written consent forms and informs the participant directly. The researcher also gives an option to the participants to not participate in the study. The participants in this study was vulnerable group (psychiatric patients/ case with mental illness), therefore, additional protection was provided"  Please review the flow of information in the methods section. From my humble point of view there are certain aspects that are not in the corresponding section (in this case ethics), besides repeating information.

Response: Thank you for your correction. This section has been moved to incorporate an ethics section.

7. The minimum number of participants recommended by Mundfrom, Shaw [29] was 20 times the number of variables, which in this case equaled 200 participants. In this study, data was missing from four questionnaires because they were not accurately completed, and 21 participants were excluded because their MMSE score was less than 24. Considering the response rate of 20%, the final sample comprised 260 participants" I seem to remember that the rule of 5 and 20 participants has long been out of use, it is calculated on the basis of expected communalities and items per factor, taking into account the work in statistical analysis under optimal conditions. I refer the authors to the citations of Ferrando-Piera et al. (2022) and Lloret-Segura et al. (2014).

Response: Thank you for your correction and suggestions. We have corrected it in accordance with your suggestions.

(Page 5)

The size of 200 participants required in this study is acceptable based on recommendations from prior studies [31, 32].

8. Sinclair, Blais [9] found that the dependability of the RSES was satisfactory, with a Cronbach's alpha of 0.91" Please determine internal consistency by scales or sub-dimensions, currently presenting internal consistency of total scale scores is a major methodological error. I refer the authors to the citation of Prinsen et al. (2018).

Response: Thank you for your correction. We have corrected and changed it in accordance with your suggestions.

(Page 6)

The validity and reliability of the RSES were assessed in prior studies, and the obtained results are presented herein. According to a previous study [33] it is suggested to ensure that the Cronbach alpha criteria for each sub-scale is ≥ 0.70. The Cronbach's alpha coefficients for the positive and negative self-esteem subscales were determined to be 0.96 and 0.98, respectively [19]. A prior study conducted on individuals who are native English speakers has also yielded Cronbach alpha coefficients of 0.87 and 0.75 for the subscales measuring positive and negative self-esteem, respectively [14]. Concerning the evaluation of validity evidence (construct validity), prior studies have demonstrated that the RSES yielded an excellent model fit [19, 34].

9. Franck, De Raedt [21] tested the construct validity by correlating the RSES score with the NEO Five-Factor Inventory subscales. In line with Tinakon and Nahathai [10], the RSES negatively correlated with the Thai Depression Inventory"  Please determine the magnitude of the correlations in the different studies. The use of some terms when mentioning types of validity evidence, do not follow the current Standards.

Response: Thank you for your correction. We have corrected and changed it in accordance with your suggestions.

(Page 6)

Concerning the evaluation of validity evidence (construct validity), prior studies have demonstrated that the RSES yielded an excellent model fit [19, 34].

10. In the last phase, the complete Indonesian version of the RSES scale was administered to the study sample, and its validity and reliability were determined"  The use of some terms when mentioning types of validity evidence, do not follow the current Standards.

Response: Thank you for your correction. We have corrected and changed it in accordance with your suggestions.

(Page 6-7)

In the final phase, the full Indonesian version of the RSES scale was administered to the study sample and tested the evidence of validity and reliability.

11. A Cronbach's alpha equal to or better than 0.70 demonstrates adequate internal consistency [33, 34]"  From my humble point of view, the authors should have calculated the ordinal alpha. The rest of the statistics in categorical response scales underestimate reliability. I refer the authors to the citation of Zumbo et al. (2007).

Response: Thank you for your correction and suggestion. In this section, we adhere to the same content as our initial composition. To the best of our knowledge, ordinal alpha is merely a measure of hypothetical reliability and does not reflect the actual reliability of the test. Herewith is an article pertaining to the misconceptions and the limited usefulness of ordinal alpha.

Reference: Chalmers, R. P. (2018). On misconceptions and the limited usefulness of ordinal alpha. Educational and Psychological Measurement, 78(6), 1056-1071. 

12. We carried out an EFA (exploratory factor analysis) to examine the construct validity of the scale using a principal component analysis [36, 37] with varimax rotation [38]" I seem to remember that the use of orthogonal rotations in highly correlated multidimensional constructs may deform the underlying structure of test item scores. I refer the authors to the citations of Ferrando-Piera et al. (2022), Lloret-Segura et al. (2014) and Prinsen et al. (2018).

Response: Thank you for your correction. After conducting a comprehensive assessment of the relevant literature, we have made the necessary revisions to our work. As a result, we have decided to exclusively focus on presenting confirmatory factor analysis (CFA) in order to demonstrate validity evidence of RSES. Therefore, EFA analysis is not performed in our revision.

13. to determine the number of components, an eigenvalue greater than 1.00 was utilized [43, 44]"  According to the latest standards it is more correct to use parallel analysis to determine the number of factors. I refer the authors to the citations of Ferrando-Piera et al. (2022) and Lloret-Segura et al. (2014). Adequacy of EFA (KMO; Barteltt's test of sphericity and determinant) should come before "parallel analysis" or "number of factors" to determine the number of factors to be retained, not after as now appears in several places of the text.

Response: Thank you for your correction. After conducting a comprehensive assessment of the relevant literature, we have made the necessary revisions to our work. As a result, we have decided to exclusively focus on presenting confirmatory factor analysis (CFA) in order to demonstrate validity evidence of RSES. Therefore, EFA analysis is not performed in our revision.

14. I do not observe the criteria used to retain or discard items in the EFA; the authors should provide scientific evidence in this regard.

Response: Thank you for your correction. After conducting a comprehensive assessment of the relevant literature, we have made the necessary revisions to our work. As a result, we have decided to exclusively focus on presenting confirmatory factor analysis (CFA) in order to demonstrate validity evidence of RSES. Therefore, EFA analysis is not performed in our revision.

15. The authors do not make it clear whether they randomly segmented the sample into two halves to perform the C

---

## [Decision Letter · Decision Letter 1]

20 Nov 2023

PONE-D-23-08354R1Construct Validity of the Rosenberg Self-Esteem Scale in patients with Schizophrenia in IndonesiaPLOS ONE

Dear Dr. Chung,

Thank you for submitting your manuscript to PLOS ONE. After careful consideration, we recommend reconsideration of your manuscript following major revision. Therefore, we invite you to submit a revised version of the manuscript that addresses the points raised during the review process.

**ACADEMIC EDITOR: Please insert comments here and delete this placeholder text when finished.** Be sure to:Please note that the discussion, as it is now, is confusing. On the one hand they state that the scale is unidimensional, but then conclude that the best structure is the two-factor model. Furthermore, in order to reach a conclusion on this, other parameters, such as factor loadings, should be taken into consideration. Factor loadings below 0.30 should not be included in these items because they do not explain enough variance. ==============================

We look forward to receiving your revised manuscript.

Kind regards,

Silvia Escribano Cubas

Academic Editor

PLOS ONE

Reviewers' comments:

Reviewer's Responses to Questions

**Comments to the Author**

1. If the authors have adequately addressed your comments raised in a previous round of review and you feel that this manuscript is now acceptable for publication, you may indicate that here to bypass the “Comments to the Author” section, enter your conflict of interest statement in the “Confidential to Editor” section, and submit your "Accept" recommendation.

Reviewer #2: (No Response)

Reviewer #3: All comments have been addressed

2. Is the manuscript technically sound, and do the data support the conclusions?

Reviewer #2: Partly

Reviewer #3: Yes

3. Has the statistical analysis been performed appropriately and rigorously? 

Reviewer #2: N/A

Reviewer #3: Yes

4. Have the authors made all data underlying the findings in their manuscript fully available?

Reviewer #2: Yes

Reviewer #3: Yes

5. Is the manuscript presented in an intelligible fashion and written in standard English?

Reviewer #2: No

Reviewer #3: Yes

6. Review Comments to the Author

Reviewer #2: I appreciate the authors' time spent in enhancing the manuscript. Statistically, it seems they have done an acceptable job. Nevertheless, there are still certain concerns I have regarding the manuscript's writing (Attached is a PDF where I've highlighted in red the elements that strike me as discordant in the text):

INTRODUCTION

1. In summary, self-esteem is a pivotal psychological construct that controls several facets of an individual's existence,

encompassing mental well-being, accomplishments, interpersonal engagements, and coping abilitiesEvidence is needed for this statement.

2. There exists a reciprocal relationship between self-esteem and mental illnesses. A previous study found that self-esteem plays a pivotal role in developing diverse mental illnesses and social problems encompassing a range of internalizing issues, such as depression, suicidal tendencies, eating disorders, and anxiety, as well as externalizing problems, including violence and substance abuse Evidence is needed for this statement. A visual representation or diagram displaying the relationship between constructs/variables would be interesting for the reader.

3. Researchers often use the RSES to measure self-esteem in the clinical population [15] Please add the characteristics of the population, diagnosis...

General Overview: The repetition of some ideas and the lack of a smooth transition between paragraphs make it difficult to read. Ensure that the citations are correctly referenced and that there is coherence in the bibliography throughout the text. Some key statements lack specific citations or references, which compromises the credibility of the text. Through a brief search, I find a large number of validations, such as those in Spanish, which the authors have not included in the relevant section of the introduction. An exposition of the structural differences and psychometric properties of the tool in the target population of the study should be made, in this case, in patients with schizophrenia.

METHODS

2.2. InstrumentsRemove the numbering from the categorical variables. Is the instruments section appropriate to discuss the known evidence of internal consistency?

2.3. Translation procedureInclude an explanatory diagram/figure of the process.

2.4 Statistical Analysis Please remove the sub-sections and write in a simpler manner.

2.4.3 Validity evidence What estimator have you used for the CFA? ULS, WLSMV?

General Overview: Please present structural validity section before reliability and internal consistency. I refer you to COSMIN. The data on internal consistency cannot be interpreted before data on structural validity.

RESULTS:

General Overview: Please provide the evidence for structural validity before internal consistency and reliability. According to COSMIN (reference provided in the first review), internal consistency and reliability cannot be interpreted without first having evidence of structural validity.

DISCUSSION

General Overview: The discussion still requires further development. Currently, the flow of information is disorganized. I also encourage the authors to delve into the results they have obtained and provide an explanation and reasoning behind their findings. They do not discuss the advantages of obtaining a single factor or the disadvantages of not obtaining it... I'm also unsure whether the studies they are comparing with have clinical or non-clinical samples. There is anthropomorphic language that should be removed. The writing needs to be revised as it still does not adhere to AERA, APA, NMCE, COSMIN standards.

Next, I copy fragments of the discussion that bother me when reading:

"Surprisingly", "can be conceptualized as comprising two distinct constructs", "In short, the evidence from our study shows that the RSES scale construct fits well", Our sample size was adequate to perform factor analysis", "Our results were consistent with a previous study by [54], conducted in", "individuals with severe mental illnesses, not specific only to patients with schizophrenia and reported strong internal consistency".......

Reviewer #3: Authors have successfully adressed all of my concerns. The manuscript can be considered for its publication now.

7. PLOS authors have the option to publish the peer review history of their article (what does this mean?). If published, this will include your full peer review and any attached files.

Reviewer #2: No

Reviewer #3: No

---

## [Author Response · Author response to Decision Letter 1]

3 Jan 2024

Dear Editor and Reviewer

Manuscript ID: PONE-D-23-08354R1

Manuscript Title: “Structural Validity of the Rosenberg Self-Esteem Scale in Patients with Schizophrenia in Indonesia”

We sincerely thank the editor and all reviewers for their valuable suggestions and for allowing us to revise our manuscript entitled “Construct Validity of the Rosenberg Self-Esteem Scale in Patients with Schizophrenia in Indonesia”. We have incorporated all the suggested changes into the manuscript and have highlighted the revised sections. At this moment, our responses and revisions are based on the editor and reviewer’s comments.

EDITORIAL COMMENTS:

Please note that the discussion, as it is now, is confusing. On the one hand they state that the scale is unidimensional, but then conclude that the best structure is the two-factor model. Furthermore, in order to reach a conclusion on this, other parameters, such as factor loadings, should be taken into consideration. Factor loadings below 0.30 should not be included in these items because they do not explain enough variance.

Response: Thank you for your correction. The necessary amendments and recommendations provided by the reviewers have been incorporated into our work. The discussion section of our study has been revised. 

(Page 10)

The adequate fit indexes were also obtained in Models 2 and 3. The two-factor and hierarchical models exhibit comparable model fit in their respective analyses. Based on the findings mentioned above, it is suggested that the RSES can be characterized as two factors, which are positive and negative self-esteem. A previous study also referred to these two factors as positive and negative self-esteem [69]. The influence of wording effect on scale items may result in or contribute to a two-factor model. The wording item effect, then further related to the method effect has been observed in earlier studies [37, 70], which suggests the presence of a two-factor of RSES [70]. In summary, our finding indicates that the RSES scale has an acceptable model fit with two factors, Similar findings were also demonstrated in a previous study that a two-factor model was deemed to have an adequate model fit [21, 36, 71, 72]. Based on our finding, we can conclude that the RSES with is a two-factor model was a valid instrument for people with schizophrenia in Indonesia. Acknowledging the necessity of reassessing the utilization of the RSES and its theoretical foundations in administering the scale to target populations is essential.

(Page 12; conclusion).

The current investigation provided evidence supporting the structural validity, internal consistency, and reliability of the RSES, indicating that the RSES can be considered a valid and reliable measurement. A two-factor model of RSES was an appropriate model to measure self-esteem in our study. This finding suggests that the use of the RSES is beneficial and applicable in assessing levels of self-esteem in individuals diagnosed with schizophrenia in Indonesia.

With respect to loading factors, the initial manuscript we submitted details that our study employed loading factors ranging from 0.69 to 0.92, indicating that all loading factors exceeded 0.30. As previously mentioned in the last revision, our analysis was limited to CFA without EFA. We therefore do not report any factor loading in our recent revision. 

REVIEWER #1 COMMENTS: No comments.

 

REVIEWER #2 COMMENTS:

I appreciate the authors' time spent in enhancing the manuscript. Statistically, it seems they have done an acceptable job. Nevertheless, there are still certain concerns I have regarding the manuscript's writing (Attached is a PDF where I've highlighted in red the elements that strike me as discordant in the text):

INTRODUCTION:

1. In summary, self-esteem is a pivotal psychological construct that controls several facets of an individual's existence, encompassing mental well-being, accomplishments, interpersonal engagements, and coping abilitiesEvidence is needed for this statement.

Response: Thank you for your correction. We have corrected and replaced it following your suggestions.

(Page 3; paragraph 1)

In summary, self-esteem is a pivotal psychological construct [6] that controls several facets of an individual's existence, encompassing mental well-being, accomplishments, interpersonal engagements, and coping abilities [3, 4].

2. There exists a reciprocal relationship between self-esteem and mental illnesses. A previous study found that self-esteem plays a pivotal role in developing diverse mental illnesses and social problems encompassing a range of internalizing issues, such as depression, suicidal tendencies, eating disorders, and anxiety, as well as externalizing problems, including violence and substance abuse Evidence is needed for this statement. A visual representation or diagram displaying the relationship between constructs/variables would be interesting for the reader.

Response: Thank you for your correction and suggestion. We have corrected and replaced it following your suggestions.

(Page 3; paragraph 2)

There is a reciprocal relationship between self-esteem and mental illnesses [7].

3. Researchers often use the RSES to measure self-esteem in the clinical population [15] Please add the characteristics of the population, diagnosis...

Response: Thank you for your correction and suggestion. We have corrected and replaced it following your suggestions.

(Page 3; paragraph 3)

Researchers often use the RSES to measure self-esteem in the clinical population, such as eating disorders [18], anxiety, depression [7], attention and emotional disorder [19], schizophrenia and bipolar disorder [20]. Other studies have tested the RSES in specific people, such as ex-prisoners [21], drug users [22], and single mothers [23].

4. General Overview: The repetition of some ideas and the lack of a smooth transition between paragraphs make it difficult to read. Ensure that the citations are correctly referenced and that there is coherence in the bibliography throughout the text. Some key statements lack specific citations or references, which compromises the credibility of the text. Through a brief search, I find a large number of validations, such as those in Spanish, which the authors have not included in the relevant section of the introduction. An exposition of the structural differences and psychometric properties of the tool in the target population of the study should be made, in this case, in patients with schizophrenia.

Response: Thank you for your correction and suggestion. We have corrected and replaced it following your suggestions, including the substitution of multiple sentences in the introduction to make it easier for readers to fully understand our idea. (See page 3-4; paragraph 1-5)

(Page 3-4; paragraph 3)

The RSES has been translated and adapted into a number of different languages, including German [24], Dutch [25], Estonian [26], French [27], Portuguese [28], Spanish [29], Japanese [17], and Thai [14]; thus, making it applicable to participants from diverse samples or populations.

METHODS:

5. InstrumentsRemove the numbering from the categorical variables. Is the instruments section appropriate to discuss the known evidence of internal consistency?

Response: Thank you for your correction and suggestion. We have corrected and replaced it following your suggestions. This section aims to elucidate the rationale for the utilization of this questionnaire. It will accomplish this by providing information about the questionnaire's developer, the scale employed, its interpretation, and the questionnaire's validity and reliability evidence, as established by prior research.

(Page 5)

The remaining variables as categorical variables, namely gender (male; female), marital status (single; married; divorced, or widowed), employment status (employed; unemployed), source of income (personal income; family support; personal and family support), education (elementary; junior; high school; university/ college), previous hospitalization (yes; no), and onset of illness (<1 year; 1-5 years; >5 years).

6. Translation procedureInclude an explanatory diagram/figure of the process.

Response: Thank you for your correction and suggestion. We have provided the flow diagram of the translation procedure (see: Figure 1)

(Figure 1)

7. Statistical Analysis Please remove the sub-sections and write in a simpler manner.

Response: Thank you for your correction and suggestion. We have corrected and replaced it following your suggestions. (Page 7-8).

8. Validity evidence What estimator have you used for the CFA? ULS, WLSMV?

Response: Thank you for your correction and suggestion. Unweighted Least Squares (ULS) was used in our analysis, since our data meet the assumption as a continuous and normally distributed.

9. General Overview: Please present structural validity section before reliability and internal consistency. I refer you to COSMIN. The data on internal consistency cannot be interpreted before data on structural validity.

Response: Thank you for your correction and suggestion. We have corrected and replaced it following your suggestions, as mentioned based on the COSMIN.

RESULTS:

10. General Overview: Please provide the evidence for structural validity before internal consistency and reliability. According to COSMIN (reference provided in the first review), internal consistency and reliability cannot be interpreted without first having evidence of structural validity.

Response: Thank you for your correction and suggestion. We have corrected and replaced it following your suggestions. 

(Page 9)

Structural validity.

The goodness of indices for all alternative models is shown in Table 3. Considering the single-factor or uni-dimensional model (M1a), the overall fit criteria were inadequate (See Figure 2). However, after some modification indices (See Figure 3), all fit criteria were significantly adequate (M1b). The adequacy of all fit criteria remained satisfactory when the two-factor model (M2) and hierarchical model (M3) were applied (See Figures 4 and 5). The AVE values were 0.69 and 0.68, and the square roots of the AVE were 0.83 and 0.82, indicating that each measured variable was significant (Table 4).

Internal consistency and reliability evidence.

The RSES had an alpha coefficient of 0.75, according to Cronbach's method. The results of Cronbach's alpha, which measures internal consistency, came in at 0.89 and 0.88 for each subscale (factor), indicating acceptable internal consistency. As presented in Table 4, the CR was calculated for positive and negative factors, and the values were 0.92 and 0.91, respectively. Test-retest reliability exhibited satisfactory results, with an ICC between 0.87 and 0.93 (Table 4).

DISCUSSION:

11. General Overview: The discussion still requires further development. Currently, the flow of information is disorganized. I also encourage the authors to delve into the results they have obtained and provide an explanation and reasoning behind their findings. They do not discuss the advantages of obtaining a single factor or the disadvantages of not obtaining it... I'm also unsure whether the studies they are comparing with have clinical or non-clinical samples. There is an anthropomorphic language that should be removed. The writing needs to be revised as it still does not adhere to AERA, APA, NMCE, COSMIN standards. 

Response: Thank you for your correction and suggestion. We have corrected and replaced it following your suggestions. Such as; construct validity was replaced specifically as structural validity; and reliability was replaced and divided as internal consistency and reliability. The details are on pages 9-11. 

Next, I copy fragments of the discussion that bother me when reading:

"Surprisingly", "can be conceptualized as comprising two distinct constructs", "In short, the evidence from our study shows that the RSES scale construct fits well", Our sample size was adequate to perform factor analysis", "Our results were consistent with a previous study by [54], conducted in", "individuals with severe mental illnesses, not specific only to patients with schizophrenia and reported strong internal consistency".......

Response: Thank you for your correction and suggestion. We have also improved the discussion section to make it easier for readers to understand what we mean in our study. The details are in the discussion section (pages 9-11). Furthermore, this manuscript was edited by Wallace Academic Editing Company to help us in providing writing corrections to our manuscript so that it complies with journal standards.

REVIEWER #3 COMMENTS:

1. Authors have successfully adressed all of my concerns. The manuscript can be considered for its publication now.

Response: We express our gratitude for your kind attention.

---

## [Decision Letter · Decision Letter 2]

9 Jan 2024

PONE-D-23-08354R2Structural Validity of the Rosenberg Self-Esteem Scale in Patients with Schizophrenia in IndonesiaPLOS ONE

Dear Dr. Chung,

Thank you for submitting your manuscript to PLOS ONE. After careful consideration, we feel that it has merit but does not fully meet PLOS ONE’s publication criteria as it currently stands. Therefore, we invite you to submit a revised version of the manuscript that addresses the points raised during the review process.

We look forward to receiving your revised manuscript.

Kind regards,

Silvia Escribano Cubas

Academic Editor

PLOS ONE

Journal Requirements:

Reviewers' comments:

Reviewer's Responses to Questions

**Comments to the Author**

1. If the authors have adequately addressed your comments raised in a previous round of review and you feel that this manuscript is now acceptable for publication, you may indicate that here to bypass the “Comments to the Author” section, enter your conflict of interest statement in the “Confidential to Editor” section, and submit your "Accept" recommendation.

Reviewer #2: All comments have been addressed

2. Is the manuscript technically sound, and do the data support the conclusions?

Reviewer #2: Yes

3. Has the statistical analysis been performed appropriately and rigorously? 

Reviewer #2: N/A

4. Have the authors made all data underlying the findings in their manuscript fully available?

Reviewer #2: Yes

5. Is the manuscript presented in an intelligible fashion and written in standard English?

Reviewer #2: Yes

6. Review Comments to the Author

Reviewer #2: In the attached document I make some recommendations to improve the scientific quality of the article.

The authors argued in previous reviews that the use of ordinal alpha was inadequate, using a critique by Chalmers. I provide a reference for you to reflect on and assess whether it would be relevant to modify the internal consistency estimator.

Zumbo, B. D., & Kroc, E. (2019). A Measurement Is a Choice and Stevens' Scales of Measurement Do Not Help Make It: A Response to Chalmers. Educational and psychological measurement, 79(6), 1184–1197. https://doi.org/10.1177/0013164419844305

7. PLOS authors have the option to publish the peer review history of their article (what does this mean?). If published, this will include your full peer review and any attached files.

Reviewer #2: No

---

## [Author Response · Author response to Decision Letter 2]

22 Feb 2024

Dear Editor and Reviewer

Manuscript ID: 

Manuscript Title: “Structural Validity of the Rosenberg Self-Esteem Scale in Patients with Schizophrenia in Indonesia”.

We sincerely thank the editor and all reviewers for their valuable suggestions and for allowing us to revise our manuscript entitled “Structural Validity of the Rosenberg Self-Esteem Scale in Patients with Schizophrenia in Indonesia”. We have incorporated all the suggested changes into the manuscript and have highlighted the revised sections. At this moment, our responses and revisions are based on the editor and reviewer’s comments.

EDITORIAL COMMENTS:

Please review your reference list to ensure that it is complete and correct. If you have cited papers that have been retracted, please include the rationale for doing so in the manuscript text, or remove these references and replace them with relevant current references.

Response: Thank you for your suggestion. We have verified the references in our manuscript and confirmed that the cited papers are still relevant.

REVIEWER #2 COMMENTS:

In the attached document I make some recommendations to improve the scientific quality of the article. The authors argued in previous reviews that the use of ordinal alpha was inadequate, using a critique by Chalmers. I provide a reference for you to reflect on and assess whether it would be relevant to modify the internal consistency estimator.

Zumbo, B. D., & Kroc, E. (2019). A Measurement Is a Choice and Stevens' Scales of Measurement Do Not Help Make It: A Response to Chalmers. Educational and psychological measurement, 79(6), 1184–1197. https://doi.org/10.1177/0013164419844305

Response: Thank you for your correction. We have corrected and replaced it following your suggestions. As a consideration, both Cronbach’s and Ordinal alpha methods produce similar results of reliability alpha.

(Page 7)

This study utilized SPSS and AMOS version 23.0 software (IBM; Armonk, New York, USA) and R Studio version 4.3.2 (R Foundation, Vienna, Austria).

(Page 8)

Ordinal coefficient alpha is considered a viable alternative coefficient alpha to calculate a reliability estimate using the Likert response data [59, 60].

(Page 9)

The overall score of the RSES has an alpha coefficient of 0.75, as determined by the ordinal alpha approach. The ordinal alpha results for each subscale were 0.89 and 0.88, demonstrating acceptable internal consistency.

---

## [Editor Report · Decision Letter 3]

23 Feb 2024

Structural Validity of the Rosenberg Self-Esteem Scale in Patients with Schizophrenia in Indonesia

PONE-D-23-08354R3

Dear Dr. Chung,

We’re pleased to inform you that your manuscript has been judged scientifically suitable for publication and will be formally accepted for publication once it meets all outstanding technical requirements.

Kind regards,

Silvia Escribano Cubas

Academic Editor

PLOS ONE